# Phenotypic Characterization by Single-Cell Mass Cytometry of Human Intrahepatic and Peripheral NK Cells in Patients with Hepatocellular Carcinoma

**DOI:** 10.3390/cells10061495

**Published:** 2021-06-14

**Authors:** Yuichi Yoshida, Sachiyo Yoshio, Taiji Yamazoe, Taizo Mori, Yuriko Tsustui, Hironari Kawai, Shiori Yoshikawa, Takasuke Fukuhara, Toru Okamoto, Yoshihiro Ono, Yu Takahashi, Ryuki Hashida, Takumi Kawaguchi, Akinobu Taketomi, Tatsuya Kanto

**Affiliations:** 1Department of Liver Diseases, The Research Center for Hepatitis and Immunology, National Center for Global Health and Medicine, Ichikawa, Chiba 272-8516, Japan; yuichi.youthfuldays@gmail.com (Y.Y.); lb-19yamazoe@hospk.ncgm.go.jp (T.Y.); lbtmori@hospk.ncgm.go.jp (T.M.); lb-19tsutsui@hospk.ncgm.go.jp (Y.T.); lb-kawai@hospk.ncgm.go.jp (H.K.); lb-19yoshikawa@hospk.ncgm.go.jp (S.Y.); 2Department of Gastoenterological Surgery I, Hokkaido University Graduate School of Medicine, Sapporo, Hokkaido 060-8648, Japan; taketomi@med.hokudai.ac.jp; 3Department of Microbiology and Immunology, Hokkaido University Graduate School of Medicine, Sapporo, Hokkaido 060-8638, Japan; fukut@pop.med.hokudai.ac.jp; 4Institute for Advanced Co-Creation Studies, Research Institute for Microbial Diseases, Osaka University, Suita, Osaka 565-0871, Japan; toru@biken.osaka-u.ac.jp; 5Division of Hepatobiliary and Pancreatic Surgery, Cancer Institute Hospital, Japanese Foundation for Cancer Research, Tokyo 135-8550, Japan; yoshihiro.ono@jfcr.or.jp (Y.O.); yu.takahashi@jfcr.or.jp (Y.T.); 6Department of Orthopedics, School of Medicine, Kurume University, Kurume 830-0011, Japan; hashida_ryuuki@med.kurume-u.ac.jp; 7Division of Gastroenterology, Kurume University School of Medicine, Kurume 830-0011, Japan; takumi@med.kurume-u.ac.jp

**Keywords:** NK cells, HCC, mass cytometory, intrahepatic lymphocytes, Siglec-10, CD160, CD49a, 2B4

## Abstract

Overall response rates of systemic therapies against advanced hepatocellular carcinoma (HCC) remain unsatisfactory. Thus, searching for new immunotherapy targets is indispensable. NK cells are crucial effectors and regulators in the tumor microenvironment and a determinant of responsiveness to checkpoint inhibitors. We revealed the landscape of NK cell phenotypes in HCC patients to find potential immunotherapy targets. Using single cell mass cytometry, we analyzed 32 surface markers on CD56^dim^ and CD56^bright^ NK cells, which included Sialic acid-binding immunoglobulin-type lectins (Siglecs). We compared peripheral NK cells between HCC patients and healthy volunteers. We also compared NK cells, in terms of their localizations, on an individual patient bases between peripheral and intrahepatic NK cells from cancerous and noncancerous liver tissues. In the HCC patient periphery, CD160^+^CD56^dim^ NK cells that expressed Siglec-7, NKp46, and NKp30 were reduced, while CD49a^+^CD56^dim^ NK cells that expressed Siglec-10 were increased. CD160 and CD49a on CD56^dim^ NK cells were significantly correlated to other NK-related markers in HCC patients, which suggested that CD160 and CD49a were signature molecules. CD49a^+^ CX3CR1^+^ Siglec-10^+^ NK cells had accumulated in HCC tissues. Considering further functional analyses, CD160, CD49a, CX3CR1, and Siglec-10 on CD56^dim^ NK cells may be targets for immunotherapies of HCC patients.

## 1. Introduction

Hepatocellular carcinoma (HCC) accounts for approximately 90% of primary liver cancers is the third leading cause of cancer deaths and is the fourth most common cancer worldwide [1,2]. The recurrence rate of HCC is high even after curative operations, because HCC usually develops in a background of chronic liver diseases that lead to cirrhosis, such as viral hepatitis, alcoholic liver disease, and non-alcoholic steatohepatitis. For early-stage HCC patients with no history of cirrhosis or portal hypertension, resection is recommended, but it is associated with recurrence rates of 60–70% at 5 years [3], which requires repeated resection and further treatment. For almost a decade, sorafenib (multi-kinase inhibitor) was the only first-line systemic targeted drug available for advanced HCC patients with a survival benefit of only 3 months. Treatments for advanced HCC patients have considerably improved over the last few years, which include immune checkpoint inhibitors (anti-PD-1 and anti-PD-L1) [4]. However, the response rates of systemic therapies are unsatisfactory. Searching for new therapeutic targets of immunological therapies is still required to improve the prognosis of patients with advanced HCC. NK cells play a critical role in regulating immune responses against tumors [5,6] and define responsiveness to checkpoint therapies in various tumor microenvironments [7]. Therefore, exploring the landscape of NK cells in HCC patients may provide a novel framework for possible immunomodulatory strategies.

Human NK cells are classified into two subsets in accordance with their expression of CD56 and CD16 (FcγRIIIa). CD56^dim^ CD16^+^ NK cells exhibit high cytotoxic activity and account for 90–95% of the peripheral NK cell population, while CD56^bright^ CD16^−^ NK cells are potent cytokine producers but are relatively scarce. In the periphery, NK cells comprise 10% of PBMCs. Conversely, NK cells comprise 30–50% of intrahepatic lymphocytes [8]. Maturation of human NK cells is characterized by loss of CD34 and C-KIT (CD117) expression, followed by sequential upregulation of CD94, CD16, and killer cell Ig-like receptors (KIRs) [9], where expression of the latter three receptors distinguishes NK cells from other members of the innate-like lymphocyte family. The functions of NK cells are regulated by HLA class I-specific inhibitory receptors (KIRs and CD94) and numerous non-HLA-specific receptors. Representative non-HLA-specific activation markers are NCRs (NKp30, NKp44, and NKp46), NKG2C, NKG2D, CD160, DNAM-1, and 2B4 [10]. Representative inhibitory receptors are NKG2A, ILT2, PD-1, PD-L1, TIGIT, Tim-3, and CD96. Sialic acid-binding immunoglobulin-like lectins (Siglecs) are a family of lectins that act as cell surface receptors for molecules that contain characteristic sialic acid linkages and transmit inhibitory signals into the cell. NK cells mainly express Siglec-7 and -9 [11]. A dichotomous relationship between the expression and functions of such Siglecs has been reported. Binding of specific antibodies or ligands to Siglec-7 and -9 inhibits NK cell functions, whereas their downregulation is indicative of NK cell dysfunction [12,13].

There are accumulated findings regarding the frequency and functions of NK cells in either the periphery or liver of patients with HCC. Most studies have reported reductions in the number and functions of NK cells in HCC patients [14,15], which are associated with their prognosis [16,17]. The reported phenotypic alterations in HCC patients are complex partly because of the dynamic changes of activating and inhibitory receptors of NK cells in accordance with their localizations. Decreases in the expression of Siglec-7, NKp46, and NKp30, and increased expression of PD-1 have been reported in the periphery [18,19]. Conversely, in the liver, decreased expression of CD160 and increased expression of CD49a, Siglec-10, PD-1, CD96, and NKG2A on intratumor NK cells have been reported [18,20,21,22,23].

In this study, we revealed the landscape of NK cell phenotypes in patients with HCC to find potential immunotherapy targets. To comprehend the relationship among multiple parameters of NK cells, we use mass cytometry to analyze NK cells among PBMCs and intrahepatic lymphocytes (IHLs). We were able to analyze the alteration and colocalization of 32 molecules on peripheral and intratumor NK cells of HCC patients, which included Siglecs and immune checkpoint molecules. We successfully identified a novel subpopulation of NK cells among PBMCs and in HCC tissues of patients, which consist of CD49a^+^CD56^dim^ NK cells with enhanced expression of CX3CR1, Siglec-10, ILT2, and PD-1. We also found that CD160 and CD49a were surrogate markers of activating and inhibitory phenotypes of peripheral CD56^dim^ NK cells, respectively, in HCC patients.

## 2. Materials and Methods

### 2.1. Study Cohorts

For evaluation of the frequencies of various immune cells, peripheral blood samples were obtained from 39 HCC patients with liver cirrhosis, who were treated with hepatic arterial infusion chemotherapy (TACE) at Kurume University Hospital from March 2019 to March 2020 (Appendix A). As controls, peripheral blood was collected from eight healthy volunteers (HVs) who had no apparent history of liver disease or malignancies and were negative for hepatitis B surface antigen, human immunodeficiency virus (HIV) antigen, anti-HIV antibodies, and anti-hepatitis C virus antibodies, who registered as volunteers at Kurume University (Appendix A). For comparison of surface markers on NK cells between HCC patients with liver cirrhosis and HVs, peripheral blood samples were obtained from 8 HCC patients with liver cirrhosis, who were admitted and treated with TACE at Kurume University Hospital from April 2020 to July 2020 (Appendix A). As controls, peripheral blood was collected from the same eight HVs as above (Appendix A). For comparison of intrahepatic NK cells from HCC tissue (Ca) and adjacent non-cancerous liver tissue (NCa), we enrolled 16 HCC patients with chronic hepatitis (*n* = 10) and liver cirrhosis (*n* = 6) who underwent liver resection for HCC at the Cancer Institute Hospital of Japanese Foundation for Cancer Research from May 2018 to November 2020 (Appendix A). For comparison of peripheral and intrahepatic NK cells, PBMCs as well as HCC and adjacent normal liver tissues were also evaluated in eight patients (Case1-8, Appendix A).

### 2.2. Isolation of Peripheral and Intrahepatic Mononuclear Cells

Peripheral blood mononuclear cells (PBMCs) were isolated by density gradient centrifugation on Ficoll-Paque (d = 1.077, Nacalai Tesque, Kyoto, Japan). To isolate intrahepatic lymphocytes (IHLs), HCC tissues and adjacent normal liver tissues were promptly transported to the laboratory from the hospital on ice in RPMI 1640 (Invitrogen, Carlsbad, CA, USA) with 2 mM L-glutamine (Gibco, Carlsbad, CA, USA), 25 mM HEPES (Nacalai Tesque), 10% FCS (HyClone, South Logan, VT, USA), and 100 U/mL penicillin/streptomycin (Nacalai Tesque) (Buffer 1). The liver tissues were washed twice with HBSS (Gibco) containning 2% FCS and 0.6% BSA (Buffer 2). The liver tissues were minced and enzymatically digested with 50 µg/L DNase I (Promega, Madison, WI, USA) and 500 mg/L collagenase IV (Nordmark Arzneimittel Gmbh & Co. KG, Uetersen, Germany) for 30–60 min at 37 °C as reported previously [8]. After the obtained cell suspension was filtered through a 40-μm cell strainer (Greiner Bio-One Inc., Frickenhausen, Germany), IHLs were isolated by density gradient centrifugation on Ficoll-Paque (*d* = 1.077, Nacalai Tesque) and CD45^+^ IHLs were obtained using the MACS system (Miltenyi Biotec, Bergisch Gladbach, Germany). PBMCs and IHLs were harvested and stored at −150 °C in Cell Banker solution (ZENOAQ RESOURCE CO., Ltd., Fukushima, Japan).

### 2.3. Mass Cytometry (Cytometry Time-of-Flight; CyTOF)

This procedure employs metal isotope-conjugated antibodies that are distinguishable by mass in a time-of-flight mass spectrometer, thereby allowing a large number of markers to be detected simultaneously without spectral overlap limitations inherent to fluorophore-based flow cytometry. For analysis, the cells were thawed, incubated with cisplatin (Fluidigm, San Francisco, CA, USA) to identify live/dead cells, and then incubated with metal-conjugated antibodies against surface membrane proteins as described below. The cells were fixed with 1.6% paraformaldehyde, labeled with an iridium-containing DNA intercalator to allow discrimination between singlets and doublets, and analyzed using a CyTOF mass cytometer (Fluidigm). CyTOF signals were normalized using EQ Beads (EQ Four Element Calibration Beads, 201078, Fluidigm) in accordance with the manufacturer’s instructions. The data files were analyzed using Cytobank software (Cytobank, Santa Clara, CA, USA). A total of 30,000 CD45^+^ leukocytes per sample was analyzed. Antibodies used for analysis are listed in Appendix A (Panel 1) and Appendix A (Panel 2). Figure 1A shows the gating strategy used to identify NK cells (CD45^+^CD3^−^CD14^−^CD19^−^CD56^+^), CD4^+^ T cells (CD45^+^CD3^+^CD4^+^CD8a^−^), CD8^+^ T cells (CD45^+^CD3^+^CD4^−^CD8a^+^), B cells (CD45^+^CD3^−^CD19^+^), T follicular helper (Tfh) cells (CD45^+^CD3^+^CD4^+^CXCR5^+^), γδT cells (CD45^+^CD3^+^CR Vdelta2^+^), MAIT cells (CD45^+^CD3^+^CD161^+^TCR Va7.2^+^), NKT cells (CD45^+^ CD3^+^TCR Va24^-^Ja18^+^), monocytes (CD45^+^CD3^−^CD14^−^CD19^+^), type 1 myeloid dendritic cells (mDC1s: CD45CD3^−^CD14^−^CD19^−^CD56^−^HLA-DR^+^CD11c^+^CD123^−^), and plasmacytoid dendritic cells (pDCs; CD45^+^CD3^−^CD19^−^CD14^−^CD56^−^HLA-DR^+^CD11c^−^CD123^+^). Data files generated from the CyTOF analysis were subjected to a dimension reduction process based on the viSNE algorithm, which allows multidimensional cytometry data to be presented in two dimensions while retaining the multidimensional data structure [24]. For viSNE analysis, 15,000 total NK cells from each donor were included and the data were clustered on the basis of the mean signal intensity (MSI).

### 2.4. Statistical Analysis

Differences between two groups were evaluated by the Mann–Whitney U-test or Wilcoxon signed-rank test. Differences between more than two groups were evaluated by the Kruskal–Wallis with Dunn’s multiple comparison test. Correlations were assessed using Spearman’s analysis. All analyses were performed using Prism software version 7 (Graph Pad, San Diego, CA, USA). Statistical analysis and data visualization were carried out using Prism software version 7 and R software packages gplots::heatmap.2 and ggplots.

## 3. Results

### 3.1. Peripheral NK Cells from Patients with HCC

#### 3.1.1. Reduced Frequency of CD56^dim^ NK Cells in HCC Patients

We first performed CyTOF to evaluate the frequencies of peripheral immune cells from eight HVs and 39 HCC patients (Appendix A). We used Panel 1 (Appendix A) and the gating strategy shown in Figure 1A. In HCC patients, NK cells, CD8^+^ T cells, MAIT cells, CD4^+^ T cells, Tfh cells, B cells, γδT cells, and pDCs were decreased, while monocytes and mDC1s were increased (Figure 1B). The frequency of CD56^dim^ NK cells was reduced in patients with HCC compared with HVs, while that of CD56^bright^ NK cells was comparable (Figure 1C).

#### 3.1.2. Decrease of CD160^+^Siglec-7^+^CD49a^-^CD56^dim^NK Cells and Increase of CD160^-^Siglec-7^-^CD49a^+^CD56^dim^NK Cells with Siglec-10, ILT2 and PD-1 Expression in the Peripheral Blood of HCC Patients

Next, we compared expression patterns of 32 surface markers on peripheral CD56^dim^ NK cells between HCC patients and HVs using Panel 2 (Appendix A). We conducted hierarchical cluster analysis to display differential expression of 32 surface markers in eight HCC patients and eight HVs (Figure 2A). A heatmap was also constructed to present the variations (Figure 2A). HCC patients and HVs were classified into different branches (Figure 2A). Representative viSNE plots of CyTOF analysis of PBMCs from an HCC patient and HV are shown in Figure 2B. We identified distinct subpopulations of CD56^dim^ NK cells. The CD56^dim^ NK subpopulation absent in HCC patients was characterized as CD160^+^Siglec-7^+^CD49a^-^ (Figure 2B, red circle). The CD56^dim^ NK subpopulation specifically present in HCC patients was characterized as CD160^-^Siglec-7^-^CD49a^+^, some of which expressed Siglec-10, ILT2, or PD-1 (Figure 2B, blue circle). Gallazzi et al. have reported that peripheral CD49a^+^CD56^bright^ NK cells, which have proinflammatory and proangiogenic features, increase in patients with prostate cancer [25]. Here, we found that peripheral CD49a^+^CD56^dim^ NK cells also increased and expressed inhibitory markers (Siglec-10, ILT2, and PD-1) in HCC patients. The expression levels and frequencies of CD160, Siglec-7, NKp46, and NKp30 on CD56^dim^ NK cells were significantly reduced in HCC patients compared with HVs (Figure 2C–E). The expression levels and frequencies of CD49a, ILT2, and PD-1 were upregulated (Figure 2C–E). Collectively, decreases of CD160, Siglec-7, NKp46 and NKp30 expression and increased expression of CD49a with ILT2 and PD-1 were distinctive features of peripheral CD56^dim^ NK cells from HCC patients. The frequency of CD160^+^CD56^dim^ NK cells was correlated positively to that of Silgec-7^+^, NKp46^+^, or NKp30^+^CD56^dim^ NK cells (Figure 2F). The frequency of CD49a^+^CD56^dim^ NK cells was correlated positively to that of Siglec-10^+^ or ILT2^+^CD56^dim^ NK cells (Figure 2F). These data suggest that CD160 and CD49a are surrogate markers of activating and inhibitory phenotypic characteristics of peripheral CD56^dim^ NK cells, respectively, in HCC patients.

#### 3.1.3. Increase of Peripheral CD49a^+^CD56^bright^NK Cells with Siglec-10 and ILT2 Expression in HCC Patients

Next, we similarly compared expression patterns of 32 surface markers on peripheral CD56^bright^ NK cells between HCC patients and HVs using Panel 2 (Appendix A). HCC patients and HVs were not classified into different branches (Appendix A). A decrease of NKp46 expression and increased expression of CD49a, Siglec-10, and ILT2 were distinctive features of peripheral CD56^bright^ NK cells from HCC patients (Appendix A). Unlike CD56^dim^ NK cells, the frequency of CD160- or Siglec-7-positive CD56^bright^ NK cells was not reduced in HCC patients (Appendix A). From the data of HVs, peripheral CD56^bright^ NK cells essentially showed lower CD160 and Siglec-7 expression levels than peripheral CD56^dim^ NK cells (Appendix A). Sun et al. have reported accumulation of intrahepatic CD49a^+^CXCR6^+^ NK cells in HCC tissues [22]. Gallazzi et al. have reported that peripheral CD49a^+^CD56^bright^ NK cells, which have proinflammatory and proangiogenic features, increase in patients with prostate cancer [25]. Here, we found that peripheral CD49a^+^CD56^bright^ NK cells were increased and expressed inhibitory markers, Siglec-10 and ILT2, in HCC patients.

### 3.2. Intratumor NK Cells in Patients with HCC

#### 3.2.1. Frequency of Intratumor CD56^dim^NK Cells Decreases in HCC Patients

First, we clarified the differences in intrahepatic NK cells from HCC tissue (Ca) and adjacent non-cancerous liver tissue (NCa) in HCC patients. We evaluated the frequencies of paired NCa-NK cells and Ca-NK cells in 16 HCC patients (Appendix A). The frequency of Ca-CD56^dim^NK cells was reduced compared with that of NCa-CD56^dim^NK cells as reported previously [14], while that of CD56^bright^ NK cells was comparable (Figure 3A).

#### 3.2.2. Increase of Intratumor CD49a^+^CD56^dim^NK cells with Siglec-10, ILT2, PD-1 and CX3CR1 in HCC Patients

Next, we compared expression patterns of 32 surface markers between Ca-CD56^dim^ NK cells and NCa-CD56^dim^ NK cells using Panel 2 in HCC patients (*n* = 16) (Appendix A). Figure 3B shows the log fold changes of MSI in Ca-CD56^dim^ NK cells with respect to that in NCa-CD56^dim^ NK cells. 2B4 expression was reduced in Ca-CD56^dim^ NK cells (3B-3E). Conversely, CD49a was highly expressed in Ca-CD56^dim^ NK cells as reported previously (3B-3E) [22]. In addition to CD49a, we found that the expression levels and frequencies of Siglec-10, ILT2, PD-1, and CX3CR1 were significantly increased in Ca-CD56^dim^ NK cells (3B-3E). The percentages of CD49a^+^Ca-CD56^dim^ NK cells correlated positively to those of Siglec-10^+^ and CX3CR1^+^ Ca-CD56^dim^ NK cells, but not ILT2^+^ or PD-1^+^ Ca-CD56^dim^ NK cells (Figure 3F), which suggested that CD49a, Siglec-10, and CX3CR1 were co-expressed in HCC tissues. NK cells express chemokine receptors CCR1, CCR2, CCR5, CXCR3, CXCR4, and CX3CR1, and migrate to inflamed sites and the tumor microenvironment [26]. Therefore, CX3CR1^+^NK cells might accumulate in the tumor microenvironment because CX3CL1 (Fractalkine), which is the ligand for CX3CR1, is produced by HCC cells [27].

Appendix A shows the log fold changes of MSI of 32 surface markers on Ca-CD56^bright^ NK cells with respect to those on NCa-CD56^bright^ NK cells. Decreased 2B4 and increased CX3CR1 expression were observed on Ca-CD56^bright^ NK cells compared with NCa-CD56^bright^ NK cells (Appendix A). The expression level and frequency of NKp30 were increased in Ca-CD56^bright^ NK cells compared with NCa-CD56^bright^ NK cells (Appendix A) as reported previously [28,29].

### 3.3. Comparison of Peripheral and Intrahepatic NK Cells in HCC Patients

We finally evaluated phenotypic differences between peripheral and intrahepatic NK cells in HCC patients. We compared 32 surface markers on peripheral NK cells, intrahepatic NCa-NK cells, and Ca-NK cells from HCC patients (*n* = 8) (Appendix A). Peripheral NK cells showed the CD160^+^Siglec-7^+^DNAM1^+^ phenotype (Figure 4A, red circle). The expression levels of CD160, Siglec-7, and DNAM-1 were higher in peripheral NK cells than in NCa-NK cells (Figure 4B,C). Intrahepatic NK cells highly expressed CD69 and NKG2D regardless of their localizations (Figure 4A–C, blue circle). CD49a and CX3CR1 expression levels were higher in intrahepatic NK cells than in peripheral NK cells, which were expressed more in HCC tissues than in non-cancerous adjacent tissues (Figure 4B,C).

## 4. Discussion

Mass cytometric analysis of immune cells enables analysis of multiple surface and intracellular molecules simultaneously at a single cell level. In this study, we performed comprehensive analysis of surface markers on peripheral and intrahepatic NK cells to explore potential new immunotherapeutic targets against HCC. We found that peripheral CD56^dim^ NK cells from HCC patients showed decreased expression of CD160, Siglec-7, NKp46, and NKp30 (Figure 2 and Figure 5). We identified a novel subpopulation of NK cells in HCC patients, which consist of CD49a^+^CD56^dim^ NK cells that co-express Siglec-10, ILT2, and PD-1 (Figure 2 and Figure 5). Such a subpopulation, CD49a^+^CD56^dim^ NK cells with Siglec-10, ILT2, and PD-1, additively express CX3CR1 more in HCC tissue compared with CD56^dim^ NK cells with the same phenotype from non-cancerous live tissues or peripheral blood (Figure 3, Figure 4 and Figure 5).

There are several advantages in identifying signature molecules of immune cells, which closely correlate to cellular functions, and hopefully, with the clinical status. In the clinical settings of HCC, such molecules, especially in PBMCs, may serve as biomarkers to monitor the occurrence and recurrence of cancer or the efficacy of therapeutic interventions. In this study, we showed that CD160 and CD49a on peripheral CD56^dim^ NK cells were candidates of such signature molecules, because each of them was significantly correlated to known activation or inhibitory molecules, respectively (Figure 2F).

CD160 is expressed specifically on NK cells with the most potent cytotoxic function [30] and is essential for IFN-γ production [31]. Decreased CD160 expression has been reported in peripheral NK cells from chronic hepatitis B patients [32] and in intratumor NK cells from HCC patients [33]. High CD160 expression on intratumor NK cells is associated with a good prognosis of HCC patients with liver resection [33]. Here, we showed that CD160 was clearly downregulated and positively correlated to activating receptors (Siglec-7, NKp46, and NKp30) in peripheral CD56^dim^ NK cells from HCC patients. HCC patients enrolled for analysis of peripheral NK cells in this study, all had liver cirrhosis (Appendix A). Thus, it is possible that the downregulation of CD160 on peripheral CD56^dim^ NK cells might result from systemic immune exhaustion due to HCC-bearing liver cirrhosis. Investigating the unknown mechanism that underlies downregulation of these activating receptors might improve the prognosis of HCC patients.

CD49a, integrin α1β1, is a marker for liver-resident NK cells (CD49a^+^CXCR6^+^NKG2C^+^CD16^−^) [34]. Moreover, intrahepatic CD49a^+^ NK cells reflect a regulatory and angiogenetic phenotype, and accumulation of CD49a^+^ NK cells in the tumor microenvironment is associated with a poor prognosis of patients with HCC [22,29]. We identified a novel subpopulation of NK cells by mass cytometry, which significantly increased in patients with HCC, the phenotypes of which was CD49a^+^CD56^dim^ NK cells with Siglec-10, ILT2, and PD-1 (Figure 2, Figure 3 and Figure 5). Such a subpopulation of NK cells was found not only in the liver, but also in the periphery of HCC patients compared with HVs (Figure 2 and Figure 5). Thus, in patients with HCC, further evaluation of the feasibility of CD160 and CD49a on peripheral NK cells as biomarkers is warranted. However, these phenotypical changes of peripheral CD56^dim^ NK cells in HCC patients with liver cirrhosis (Figure 1 and Figure 2) could be induced by liver cirrhosis as well as by HCC itself. To clarify the impact of HCC itself on peripheral CD56^dim^ NK cells, samples with a similar proportion of age and gender should be compared among healthy volunteers, chronic hepatitis patients without HCC, chronic hepatitis patients with HCC, liver cirrhosis patients without HCC and liver cirrhosis patients with HCC.

We compared the phenotypes of NK cells among different localizations in eight HCC patients, namely the periphery and in non-cancerous and cancerous tissues. We found three types of expression patterns of surface molecules on NK cells. First, the molecules expressed more in the periphery than in the liver were CD160, Siglec-7, and DNAM-1, as well as CD27, CD94, CD96, KIR2SL1, KIR3DL1, NKG2C, LAG3, and TIM-3 (Figure 4 and data not shown). Exceptionally, NKp30 was decreased in NCa-NK cells compared with peripheral NK cells, but increased in Ca-NK cells to the same levels in peripheral NK cells (data not shown). It has been reported that NK cells positive for CD160, Siglec-7, and DNAM-1 are highly functional. Thus, downregulation of such markers may reflect tolerizing pressure in the liver regardless of the presence of HCC. Second, a set of molecules was expressed more on NK cells in the liver, regardless of cancerous or non-cancerous, compared with peripheral NK cells, CD69, and NKG2D, which are highly expressed on liver resident CD56^bright^ NK cells (Figure 4). Third, the expression of CD49a and CX3CR1 on NK cells was higher in the liver than in the periphery. Siglec-10 and ILT2 also tended to be higher in the liver than in the periphery. Of particular interest, CD49a, CX3CR1, Siglec-10, and ILT2 were expressed more on cancerous NK cells than non-cancerous NK cells (Figure 3 and Figure 4). These results suggest that CD49a, CX3CR1, Siglec-10 and ILT2 might be induced by HCC. Further investigation is needed to clarify the mechanisms of induction and action of such molecules on NK cells in the tumor microenvironment in HCC.

Peripheral CD49a^+^CD56^bright^ NK cells and intratumor CD49a^+^CD56^dim^ NK cells were increased and expressed Siglec-10 in HCC patients. The interaction of Siglec-10 with its ligand CD24 on tumor cells is a mechanism of tumor immune escape [35,36]. High Siglec-10 expression on intratumor NK cells is associated with a poor prognosis of HCC patients [23]. In our HCC patients, the Siglec-10 expression level on peripheral CD56^dim^ NK cells was positively correlated to the tumor size (T stage) (Appendix A) [37]. No correlation was observed between any other surface marker and clinical parameter, which included tumor factors, liver functions, and prognosis, possibly because of the small number of patients. It is obvious that Siglecs and their ligands potentially serve as novel immune checkpoints in patients with various malignancies. Identification of yet-undisclosed molecular profiles of Siglec ligands definitely prompt us to consider the Siglec family as therapeutic targets for HCC.

In summary, we performed comprehensive analyses of the phenotypic characteristics of intrahepatic and peripheral NK cells in HCC patients. We found the decrease of CD160, Siglec-7, NKp46 and NKp30 expression and increase of CD49a, Siglec-10, PD-1 and ILT2 expression on peripheral NK cells in HCC patients with liver cirrhosis, compared with HVs (Figure 5). We also found accumulation of CD49a^+^ NK cells with CX3CR1, Siglec-10, ILT2, and PD-1 expression in HCC tissues (Figure 5). These findings merit further studies to investigate the possibility that CD160, CD49a, Siglec-10, ILT2 are new therapeutic targets or biomarkers for disease progression or the therapeutic efficacy of systemic therapies.

## Figures and Tables

**Figure 1 cells-10-01495-f001:**
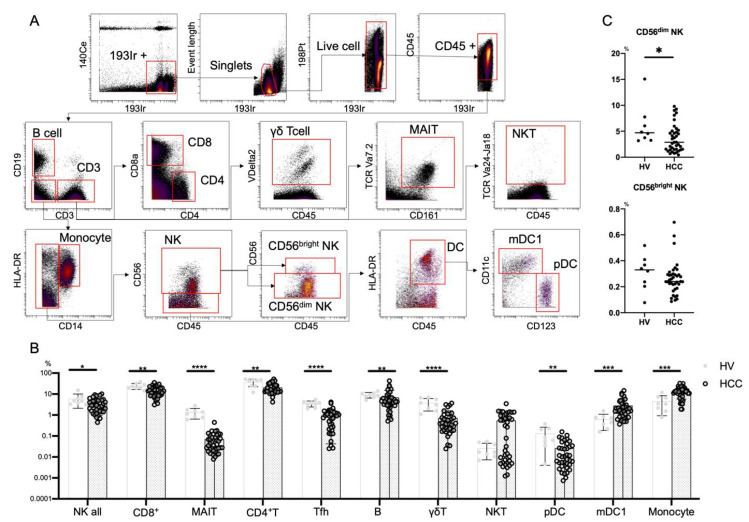
Frequencies of immune cells in HCC patients. (**A**) Mass cytometry gating scheme used to identify the peripheral blood immune cell subsets investigated here. Arrows indicate the gating sequence. (**B**) Summary of the frequency frequencies of the indicated immune cell populations isolated from healthy volunteers (HVs) (*n* = 8) and HCC patients (*n* = 39). (**C**) Percentages of CD56^dim^ NK cells and CD56^bright^ NK cells among total peripheral blood mononuclear cells from HVs (*n* = 8) and HCC patients (*n* = 39). Data are presented as means ± SD (**B**,**C**). * *p* < 0.05; ** *p* < 0.01; *** *p* < 0.001; **** *p* < 0.0001 by the Mann–Whitney U-test (**B**,**C**).

**Figure 2 cells-10-01495-f002:**
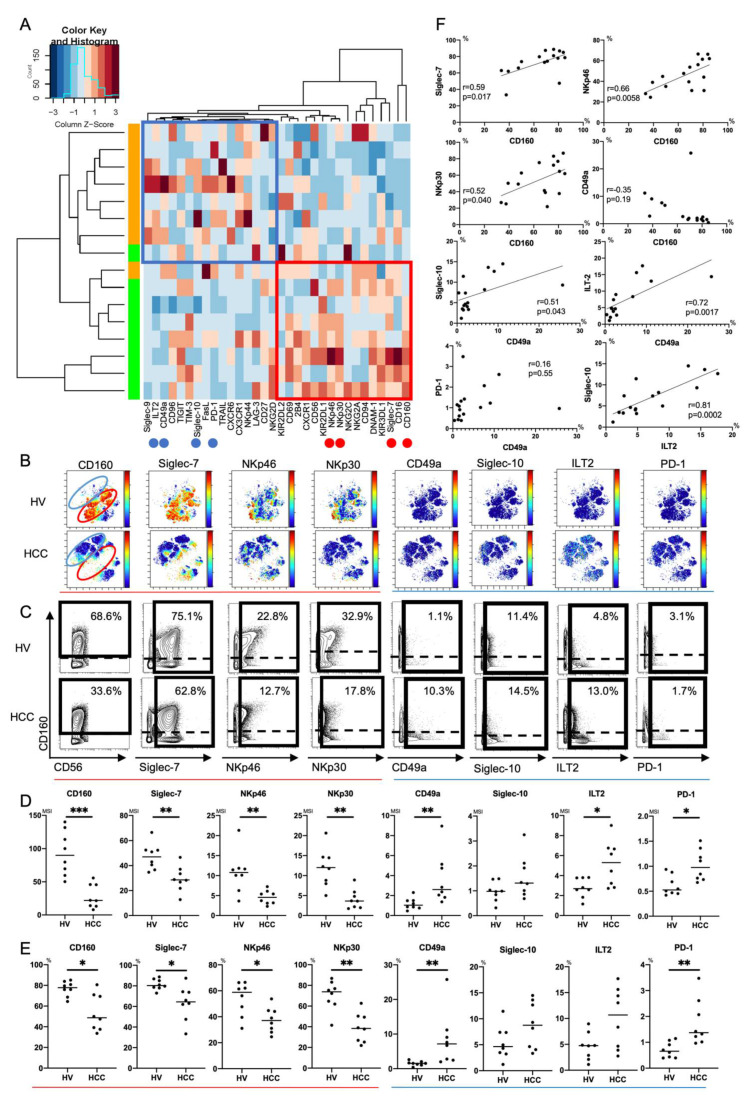
Phenotypic characterization of peripheral CD56^dim^ NK cells from HCC patients. (**A**) Heatmap and hierarchical clustering of the surface marker expression on CD56^dim^ NK cells (CD45^+^CD3^-^CD14^-^CD19^-^CD56^dim^) from HCC patients and healthy volunteers (HVs). Each expression level was scaled through samples and z scores were used for the color scale. The horizontal axis represents 32 surface markers on CD56^dim^ NK cells. The vertical axis represents samples of HCC patients and HVs. The heatmap indicates the deviation score of expression levels of surface markers in any sample. The blue region represents downregulated markers and the red region represents upregulated markers. Red circles indicate surface markers that were expressed highly on CD56^dim^ NK cells from HVs compared with those from HCC patients (CD160, Siglec-7, NKp46, and NKp30). Blue circles indicate surface markers that were expressed highly on CD56^dim^ NK cells from HCC patients compared with those from HVs (CD49a, ILT2, and PD-1). (**B**) Representative viSNE plots of peripheral CD56^dim^ NK cells from HVs and HCC patients. Each panel shows expression of the indicated surface protein (CD160, Siglec-7, NKp46, NKp30, CD49a, Siglec-10, ILT2, and PD-1). The color scale indicates a gradient of high (red) to low (blue) expression of the relevant protein. The red circle indicates the CD56^dim^ NK subpopulation that was specifically detected in HVs. The blue circle indicates the CD56^dim^ NK subpopulation that was specifically detected in HCC patients. (**C**) Representative mass cytometry plots, (**D**) quantification (median signal intensity; MSI) (**E**) percentages (%) of CD160, Siglec-7, NKp46, NKp30, CD49a, Siglec-10, ILT2, and PD-1 expression on CD56^dim^ NK cells from HCC patients (*n* = 8) and HVs (*n* = 8). Data are presented as individual values with a mean line. * *p* < 0.05; ** *p* < 0.01; *** *p* < 0.001 by the Mann–Whitney U-test. Red and blue lines indicate activating and inhibitory receptors, respectively. (**F**) Correlations between expression (MSI) of CD160, Siglec-7, NKp46, NKp30, CD49a, Sigelec-10, ILT2, and PD-1 on CD56^dim^ NK cells in HCC patients (*n* = 8) and HVs (*n* = 8). *p*-values and correlation coefficients (*r*) were calculated by Spearman’s correlation test.

**Figure 3 cells-10-01495-f003:**
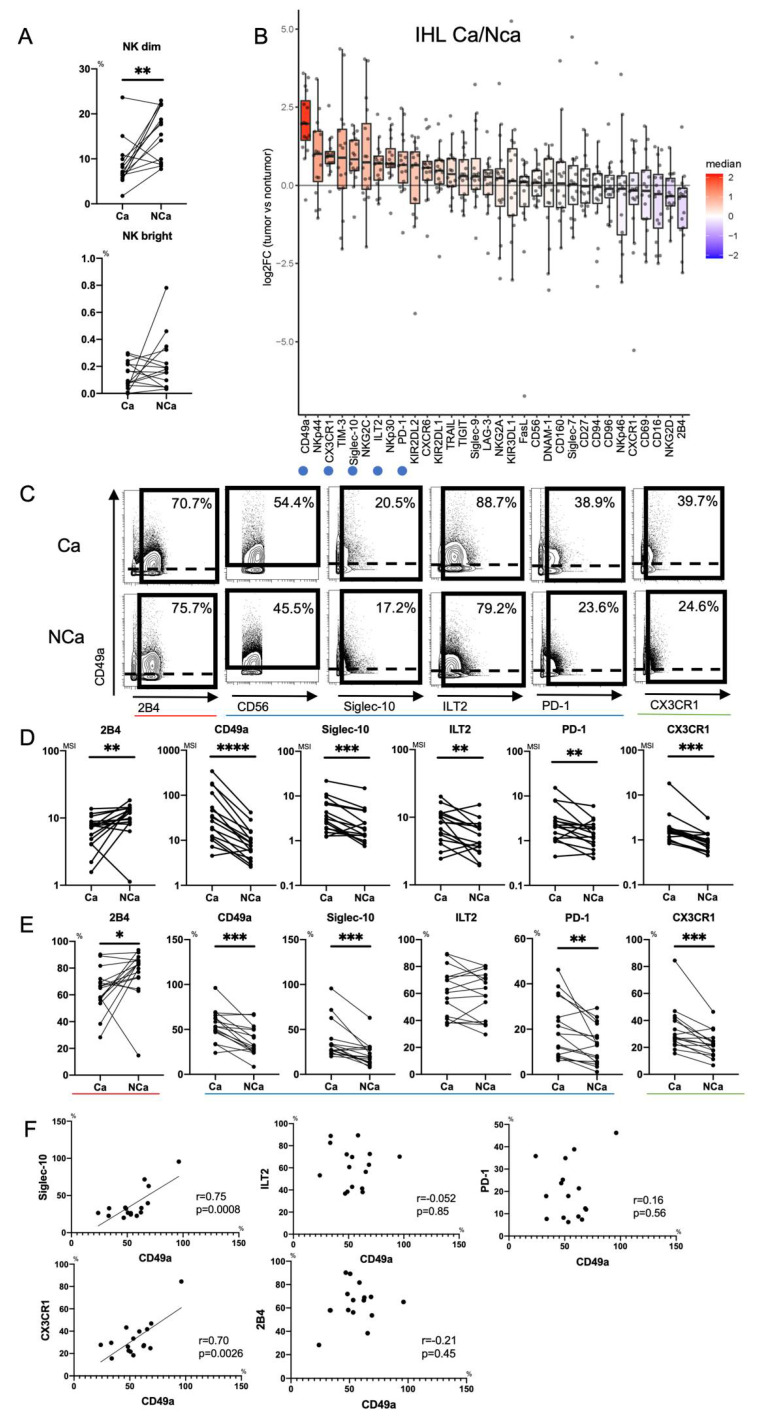
Phenotypic characterization of intrahepatic CD56^dim^ NK cells from HCC tissues. (**A**) Percentages of intrahepatic CD56^dim^ NK cells (CD45^+^CD3^−^CD14^−^CD19^−^CD56^dim^) and CD56^bright^ NK cells (CD45^+^CD3^−^CD14^−^CD19^−^CD56^bright^) from HCC tissue (Ca) and non-cancerous adjacent liver tissue (NCa) in 16 HCC patients. ** *p* < 0.01; *** *p* < 0.001 by the paired *t*-test. (**B**) Relative expression of surface markers on Ca-CD56^dim^ NK cells and NCa-CD56^dim^ NK cells (*n* = 16). The horizontal axis indicates log fold changes of MSI in Ca-CD56^dim^ NK cells with respect to NCa-CD56^dim^ NK cells. All boxplots show the median, 1.5 inter-quartile range (IQR), and upper and lower quantiles. Blue (inhibitory markers) and green (chemokine receptor) circles indicate markers that were significantly increased in Ca-CD56^dim^ NK cells. Red circle indicates the marker that was significantly reduced in Ca-CD56^dim^ NK cells. (**C**) Representative mass cytometry plots, (**D**) quantification (MSI), and (**E**) percentages (%) of CD49a, Siglec-10, ILT2, PD-1, CX3CR1, CD160, Siglec-7, NKp46, NKp30, and 2B4 expression on Ca-CD56^dim^ NK and NCa-CD56^dim^ NK cells. * *p* < 0.05; ** *p* < 0.01; *** *p* < 0.001; **** *p* < 0.0001 by the paired *t*-test. Red, blue, and green lines indicate activating, inhibitory, and chemokine receptors, respectively. (**F**) Correlation of the percentage of CD49a^+^Ca-CD56^dim^ NK cells with Siglec-10, ILT2, PD-1, and CX3CR1-positive Ca-CD56^dim^ NK cells was evaluated in 16 HCC patients. *p*-values and correlation coefficients (*r*) were calculated by Spearman’s correlation test.

**Figure 4 cells-10-01495-f004:**
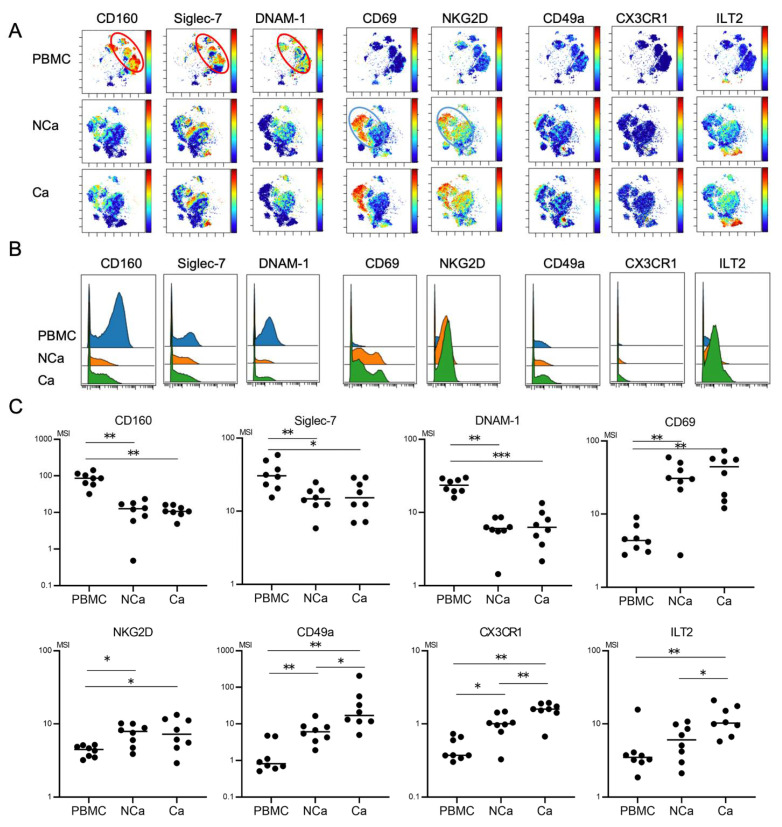
Comparison of phenotypic characterization between peripheral and intrahepatic NK cells from HCC patients. (**A**) Representative viSNE plots of peripheral and intrahepatic NK cells (CD45^+^CD3^−^CD14^−^CD19^−^CD56^+^) from HCC patients. Each panel shows expression of the indicated surface protein. The color scale indicates a gradient of high (red) to low (blue) expression of the relevant protein. PBMC, peripheral blood mononuclear cell; NCa, intrahepatic lymphocytes from non-cancerous adjacent liver tissues; Ca, intrahepatic lymphocytes from HCC tissues. (**B**) Representative histograms showing expression of the indicated cell surface protein in peripheral and intrahepatic NCa-and Ca-NK cells from HCC patients. (**C**) Quantification of CD160, Siglec-7, DNAM1, CD69, NKG2D, CD49a, CX3CR1, Siglec-10, ILT2, and PD-1 expression (MSI) on peripheral, intrahepatic NCa- and Ca-NK cells. * *p* < 0.05; ** *p* < 0.01 by the Kruskal–Wallis test with Dunn’s multiple comparison test.

**Figure 5 cells-10-01495-f005:**
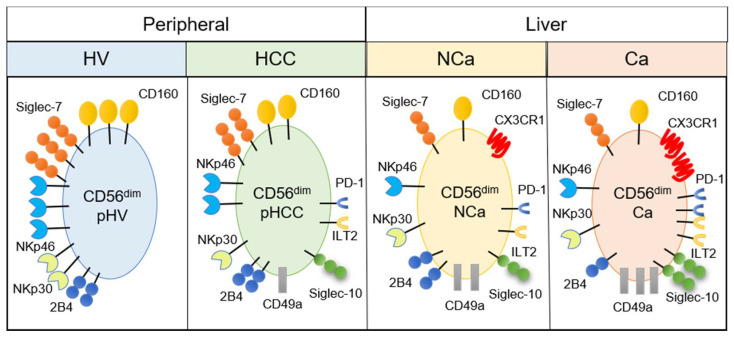
Summary of phenotypic alterations in peripheral and intrahepatic CD56^dim^ NK cells from HCC patients. CD56^dim^, CD56^dim^ NK cells; pHV, in the peripheral blood of healthy volunteers; pHCC, in the peripheral blood of HCC patients; NCa, from noncancerous adjacent liver tissues; Ca, in HCC tissues.

## Data Availability

The data presented in this study are available on request from the corresponding author.

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
