# Peer review of "Phenotypic Characterization by Single-Cell Mass Cytometry of Human Intrahepatic and Peripheral NK Cells in Patients with Hepatocellular Carcinoma"

_cells, 2021, doi:10.3390/cells10061495_

Round 1
Reviewer 1 Report
The study by Dr. Yoshida and collaborators is aimed at performing a characterization of intrahepatic and peripheral Nature Killer (NK) cell population in patients with hepatocellular carcinoma (HCC) with the final goal of finding novel potential immunotherapy targets regarding NK subpopulation.
Authors make a thorough characterization of NK cells by means of CyTOF in two different cohorts of HCC patients and a group of healthy volunteers, both in blood and tissue samples and finally identify a novel NK cell subpopulation that is increased in HCC patients.
The aim of the study is clear, and methodology is correct, but study design shows several flaws:
Major Comments:
- The most important problem relates to the design of patient cohorts. First cohort of 39 HCC patients are compared to a control group of 8 healthy individuals. Those groups are not balanced according to gender and age. While all volunteers are male, HCC group shows half their population being females. The same goes with age, 31.4 years in healthy group and 80.4 years in HCC group. Both age and gender have a well-known impact on immune system. Moreover, if authors aim to identify cell populations associated with HCC the correct approach should be to compare blood samples from HCC patients with samples from patients with liver disease (cirrhosis, chronic hepatitis and steatosis/steatohepatitis) without the evidence of liver cancer. Obviously, liver diseases shapes the phenotype of liver resident immune cell populations.
- The second group of HCC patients (n=16) analyzed in this study seems to be a completely different HCC patient cohort. First, gender balance is approximately 4:1 favoring males and mean age is lower than the first group of 39 patients. Authors should be able to guarantee that both groups of HCC patients have similar tumor characteristics and similar underlying liver disease. For instance, PBMC analysis from first (n=8) and second (n=8) HCC cohorts are they comparable?
- It would be useful to add a simple hepatic profile biochemistry from healthy volunteers in Table S1, simply to rule out any liver disease.
- Table S5 in column corresponding to T factor, we assume it correspond to TNM classification. For instance, a T3 stage should correspond to the presence of several nodules with one nodule being larger than 5cm. In this table, patient 1 is classified as T3 and single tumor (St). There are several other patients with similar inconsistencies. Authors should correct tumor characteristics or specify which tumor staging classification is using.
- The final paragraph of the discussion must be tempered. Authors state that “…..CD160 and CD49a expression reflects the activating and inhibitory status of peripheral NK cells, respectively” Although data in the literature supports the fact of CD160 being activating and CD49a inhibitory in NK cells, this study only evidences their presence but not their function.
Minor Comments:
- Line 114: Table S2does not show clinicopathological characteristics of 16 patients
- Line 210: Figure 1. Frequencies of immune cells in HCC patients. (B) graphic expressing the frequencies of different immune cell populations should be changed for a graphic combining box plots showing the data from each patient. This will evidence the distribution of cell populations for each group, HCC patients and healthy volunteers.
- Line 229: Figure 2. (C) quantification (MSI) and (D) percentages should be corrected. (C) are percentages and (D) corresponds to MSI quantification.
- Line 272: Figure 3. (C) and (D) show the same mistake than that of Figure 2
- Figure S1: Again (B) and (C) are mistaken
- Table S1 show no legend, no acronym description.
- Table S3 show no legend, no acronym description.
- Table S5 several acronyms lack the description: F stage, C (etiology column), PT.
Author Response
We appreciate for the thoughtful comments and suggestions from you. Please see our responses in the attached PDF file.

Reviewer 2 Report
The manuscript “Phenotypic characterization by single-cell mass cytometry of 2 human intrahepatic and peripheral NK cells in patients with 3 hepatocellular carcinoma” reports mass cytometry based analysis of NK cell population in HCC patients and normal volunteers from different sites: cancer, matched non-cancerous tissue and PBMCs. The authors also report a novel subpopulation of NK cells which are cd49+ and also express siglec-10, ILT2 and PD-1.
The findings in this manuscript are interesting and will be of interest to the HCC immunology community.
However, a lack of functional study weakens the impact of the findings. Are the authors planning on doing mice studies on these subsets of cells? The difference in subpopulations in HV vs HCC or HCC vs matched normal might only be a result of transformation and not have any functional role.
I have a couple of suggestions:
-Please make a comment on iNKT cells in Figure 1, no mention has been made as to how they were characterized or even what the roles are and so forth.
-In Figure 3A, NK dim percentage is decreasing in three patients; are the same patients showing high NK bright in NCa? This is not clear. Figure 3A needs to be a bit more clearly conveying the main message.
Author Response
We appreciate for the thoughtful comments and suggestions from the reviewers. Our responses to them are in the attached PDF file. Please see the attachment.

Round 2
Reviewer 1 Report
The authors revised the manuscript adequately according to reviewers’ comments. They improved the description of patient cohorts and improved the discussion by including the limitations of the study.
However, in my first revision I asked for a change in Figure 1 Panel B. I asked for a graph combining scatter plot with columns, this way it would be easier understanding the huge standard deviation values (SD) in specific cell populations. Really, Figure 1 panel D does not supply any further information. I suggest the authors consider changing the graph.
